# Nordic Seas polynyas and their role in preconditioning marine productivity during the Last Glacial Maximum

Jochen Knies[1,2], Denizcan Köseoğlu[3], Leif Rise[1], Nicole Baeten[1], Valérie K. Bellec[1], Reidulv Bøe[1], Martin Klug[1], Giuliana Panieri [2], Patrycja E. Jernas[2,4] & Simon T. Belt [3]

Arctic and Antarctic polynyas are crucial sites for deep-water formation, which helps sustain global ocean circulation. During glacial times, the occurrence of polynyas proximal to expansive ice sheets in both hemispheres has been proposed to explain limited ocean ventilation and a habitat requirement for marine and higher-trophic terrestrial fauna. Nonetheless, their existence remains equivocal, not least due to the hitherto paucity of sufficiently characteristic proxy data. Here we demonstrate polynya formation in front of the NW Eurasian ice sheets during the Last Glacial Maximum (LGM), which resulted from katabatic winds blowing seaward of the ice shelves and upwelling of warm, sub-surface Atlantic water. These polynyas sustained ice-sheet build-up, ocean ventilation, and marine productivity in an otherwise glacial Arctic desert. Following the catastrophic meltwater discharge from the collapsing ice sheets at ~17.5 ka BP, polynya formation ceased, marine productivity declined dramatically, and sea ice expanded rapidly to cover the entire Nordic Seas.

[1] Geological Survey of Norway, NO-7491, Trondheim, Norway. [2] CAGE - Centre for Arctic Gas Hydrate, Environment and Climate; Department of Geosciences, UiT The Arctic University of Norway, NO-9037, Tromsø, Norway. [3] Biogeochemistry Research Centre, School of Geography, Earth and Environmental Sciences, University of Plymouth, Plymouth PL4 8AA, UK. [4] Department of Marine Geology, Institute of Oceanography, University of Gdańsk, Al. Piłsudskiego 46, 81-378 Gdynia, Poland. Correspondence and requests for materials should be addressed to J.K. (email: jochen.knies@ngu.no)

Coastal polynyas on Arctic and Antarctic shelves today are widely recognized to be sites of deep water formation[1,2] and data from both hemispheres have illustrated their importance as a significant driver of the Atlantic meridional overturning circulation (AMOC)[3]. A major source of bottom waters is found along the East Antarctic coastline in a myriad of coastal polynyas that form in response to strong katabatic winds originating from the adjacent ice masses[4]. High rates of sea ice growth and consequential ejection of salt in these polynyas contribute to the production of dense shelf waters (brines), and thus deep water renewal in the global ocean. Although this model was proposed initially to explain persistent ocean convection during past cold phases (stadials) in the glacial Nordic Seas[5], it was subsequently questioned, since brine-enriched shelf waters are isotopically indistinguishable from deep water generated by ocean convection[6,7]. More recently, however, Keigwin and Swift[8] found evidence for glacial deep water in the western North Atlantic that may have been formed through enhanced sea ice growth rates and brine rejection in the Labrador Sea, potentially associated with polynya formation. In the Nordic Seas, Thornalley, et al.[9] also discussed the possibility of limited deep-water formation as a consequence of brine rejection from coastal polynyas. The existence and location of these polynyas, however, remain unresolved.

In Antarctica, coastal polynyas are inferred to have occurred beyond the glacial ice-sheet margin, with their formation probably enhanced by offshore katabatic winds and amplified further by diurnal tides during the last glacial period and the Holocene[10–12]. The presence of the grounded ice sheet at the shelf edge maintained the formation of coastal polynyas between ca. 25 and 19 ka[13], which then became inactive when the ice sheet retreated. Apart from their importance for deep water formation, coastal polynyas adjacent to expansive ice sheets are also regional moisture sources[11,14–16]. Further, they are recognized as key sites for enhanced primary and secondary productivity associated with sea-ice controlled seasonal nutrient supply during advance and retreat, and thus represent oases for higher-trophic life in an otherwise glacial desert[17].

Corroboration of coastal polynyas as suppliers of moisture to the build-up of adjacent ice sheets, and as potential sea-ice factories that facilitate regional ventilation and primary productivity in the glacial Nordic Seas, requires confirmation of their existence through multi-proxy evidence preserved in the sedimentary archive. Since the Barents Sea shelf was fully glaciated during the Last Glacial Maximum (LGM) between 26.5 and 19 ka[18–20], the margins were preconditioned for the formation of coastal polynyas. Previously, the dominance of planktic and benthic foraminifera and the occurrence of the seasonal sea ice biomarker IP$_{25}$ in LGM sediments from the northern Barents Sea margin have been used to propose the katabatic wind-driven formation of coastal polynyas supported by upwelling of Atlantic-derived water masses[15,16]. Meanwhile, at the western margin, advection of Atlantic water (AW) and seasonally sea ice free conditions prevailed during the LGM[21,22]. Although no direct inferences of polynya formation have been made so far in this region, Bauch, et al.[14] postulated that sub-surface Atlantic water advection into the Nordic Seas was possibly facilitated by polynyas.

In the present study, we combine multi-biological proxy data with sedimentary physico-chemical characteristics to show that coastal polynyas indeed existed along the entire Svalbard-Barents Sea margin during the LGM, based on the evidence of a highly dynamic sea ice cover coupled with enhanced plankton productivity adjacent to a grounded ice sheet at the continental shelf. In contrast to previous studies, we propose the existence of an open water corridor in front of the NW Eurasian ice sheet that was controlled, primarily, by a combination of strong katabatic

winds blowing seaward off the ice shelves and upwelling of relatively warm intermediate Atlantic water masses. This polynya activity along the entire continental margin of the Barents Sea was the ultimate reason for weak (but constant) ocean convection and persistence of higher-trophic life in otherwise heavily sea-ice covered Nordic Seas during the LGM. The duration of this polynya scenario is closely tied to the stability of the marine-based Svalbard-Barents Sea ice sheet (SBIS). Thus, with the onset of the SBIS deglaciation at ~19.5 ka, biological activity in the polynyas ceased, and finally stopped when the ice sheet collapsed at ~17.5 ka, triggering the formation of perennial sea ice cover over the entire Nordic Seas as a consequence of extreme freshwater modulation of the surface ocean and significant weakening of the AMOC.

## Results

**Physiogeography and geological setting.** The study area (Fig. 1) was mapped using multibeam echosounder (Kongsberg Simrad EM 710, 70-100 kHz range) by the Norwegian Mapping Authority (NMA) and the Norwegian Defence Research Establishment (FFI) in 2008–2009 as part of the Norwegian offshore seabed mapping program MAREANO (www.mareano.no)[23]. During two cruises with RV *G.O. Sars* in 2012 and 2014, a gravity core (33-GC08; hereafter GC08) and a giant piston core (GS14-190-01PC; hereafter referred to as GS14-190) were retrieved from the same location on the upper continental slope in ~949 m water depth (Fig. 1, Supplementary Table 1). While sediments from both cores are used to establish the chronological framework, paleoenvironmental inferences are made from piston core GS14-190 only. In this study, we present data from the upper 20–700 cm core depth (0–20 cm interval was not properly recovered), representing ca. 15 to 32 ka BP. Fig. 1 illustrates the glacial character of the SW Barents Sea shelf. It has been glaciated multiple times during the Quaternary[24,25]. During the LGM, SBIS waxed and waned over the shelf, with major ice streams that operated in cross-shelf troughs, including the Bear Island Trough (Fig. 1)[20,26,27]. Discharge of large volumes of sediment and meltwater occurred during the SBIS collapse along with associated climate and ocean warming. As a consequence, the continental slope in the southwestern Barents Sea is characterized by multiple gullies that document the prevalence of glacigenic debris flows and mass-movement activity during glacials, and meltwater discharge during deglacial periods (Fig. 1)[23,28]. The modern oceanographic setting is dominated by the Norwegian Atlantic Current (NAC), which transports water of Atlantic origin northward through the Norwegian Sea[29]. North Atlantic Water is characterized by high salinity (>35‰) and temperature (>6 °C) at the mid-Norwegian margin, and is typically found on the upper slope shallower than ~700 m water depth[29,30].

**Lithology, chronology and sedimentation rates.** Visual inspection, X-ray photography, grain size analyses and physical properties have been used to describe the sediments of GS14-190 (Fig. 2). We have identified two types of sediment facies: Facies (1) Hemipelagic, silty-clay mud intercalated with ice-rafted debris (IRD) or dropstones with occasional signs of lamination. Silty, clay-rich sediments have wet bulk density <2 g/cm³, water content between 40 and 60%, and low magnetic susceptibility (MS ~50 $10^{-5}$ SI). Sequences enriched in IRD are slightly denser (~2.1 g/cm³) and/or characterized by elevated MS values. Facies (2) This facies is more heterogenous with higher amounts of coarse-sand, higher density (2.2 g/cm³) and lower water content (~25–30%). The two short intervals around 442 and 540 cm with Facies 2 characteristics are interpreted as mass flow deposits. The core top (ca. 20–30 core depth cm) is sandy-rich and most likely a

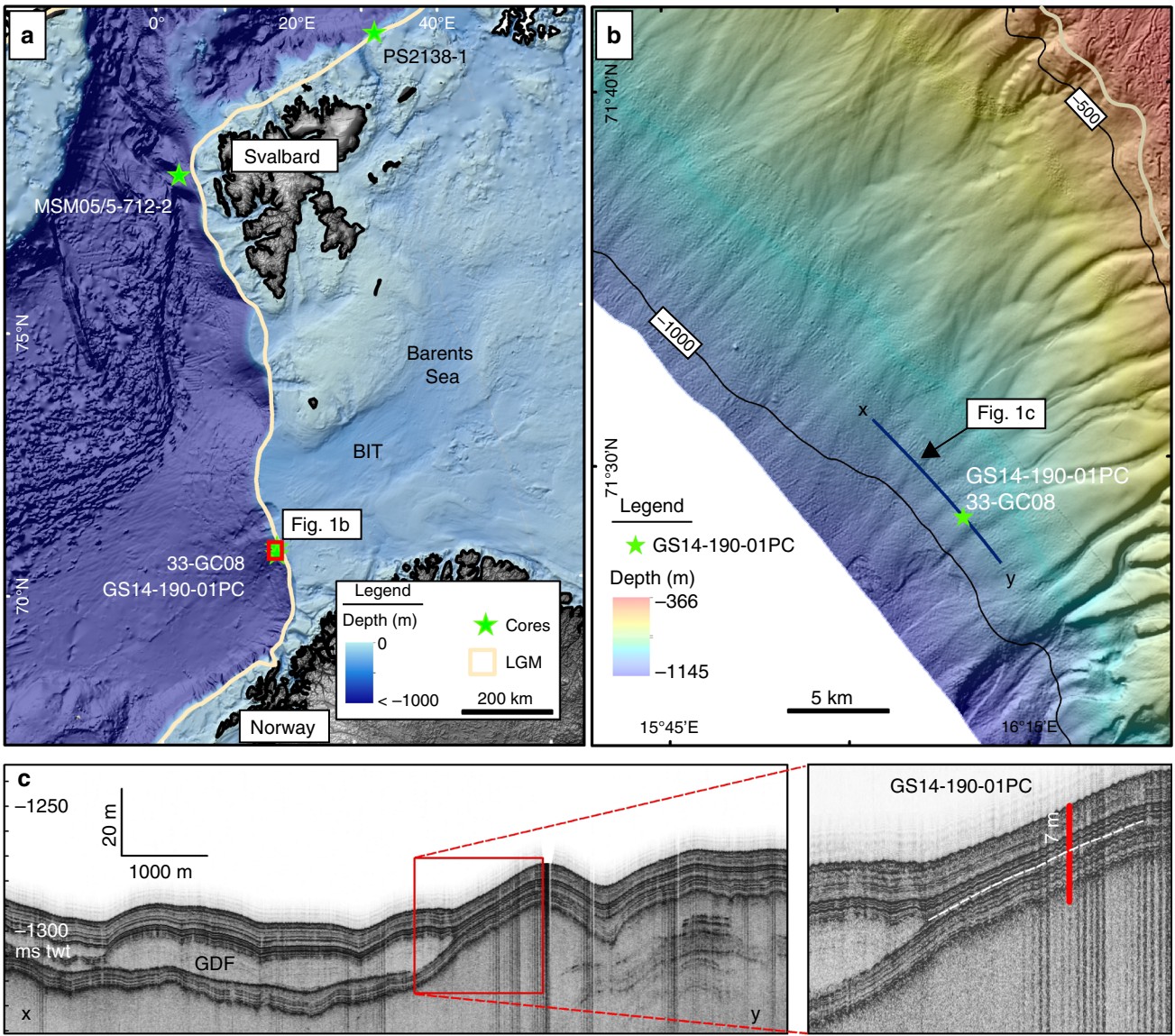

**Fig. 1** Study area. **a** Present physiogeography with discussed sediment core locations. The outline of maximum extent of the Late Weichselian ice sheet is indicated[18]. **b** Site location with detailed bathymetry and shaded relief map with core positions GS14-190-01PC/33-GC08 and available seismic (TOPAS) line shown in **c**. Distinct morphological features of the slope are visible. Maximum ice sheet limit is shown (beige line). **c** High-resolution seismic data (TOPAS). Glacial Debris Flows (GDF) are interbedded with glaciomarine sediments. GDFs are characterized by acoustically uniform, transparent seismic signatures. Inset (red rectangle) shows position of piston core GS14-190-01PC penetrating the package of acoustically stratified sediments. The upper 700 cm of the core penetration is marked (stippled white line)

distal deposit of nearby reported sandwaves[23] (Fig. 2). This part of the core has not been sampled for this study, however.

The chronology of the upper section (20–700 cm) of GS14-190 is constrained by 6 accelerator mass spectrometry (AMS) radiocarbon datings obtained from tests of planktic or mixed benthic and planktic foraminifera (Table 1). The age model is further supported by 6 AMS radiocarbon datings of bivalve shells (*Thyasira* sp.) and mixed benthic and planktic foraminiferal assemblages from parallel core GC08 (Table 1). All AMS $^{14}$C dates were calibrated to calendar ages (cal. kyr BP) by applying the Calib7.1 program and the Marine13 calibration curve[31]. No local reservoir age ($\Delta R = 0$) was applied. We are aware of large uncertainties in reservoir ages in the Nordic Seas during the LGM and early deglaciation[9,32] and the consequence of younger ages when applying local reservoir ages between $\Delta R = 0$ to 400–500. Here, we elected to use $\Delta R = 0$ to enable us to make a direct comparison to other key records from the North Atlantic and

Nordic Seas. To generate a common depth model for both cores, we used the XRF Ca records to identify several tie-points (0–5) (Fig. 3, Table 2) using the AnalySeries software[33], thus permitting the transfer of AMS $^{14}$C dates in GC08 into the age model of GS14-190 (Table 3). The age-depth model for core GS14-190 was generated using the Bayesian age-modeling (Bacon) approach on 12 AMS $^{14}$C dates (Bacon v2.2)[34] (Fig. 4).

**Sedimentological and geochemical proxies.** The stratigraphic framework of GS14-190 is supported further by the $\delta^{18}O$ record of planktic foraminifera *Neogloboquadrina pachyderma* sinistral (sin.). The LGM is characterized by the heaviest $\delta^{18}O$ values (ranging from 4.9‰ to 5.2‰) and rapid $\delta^{18}O$ depletion during the deglacial phase at ~17.5 ka BP (Fig. 5). The seasonal sea ice proxy IP$_{25}$ is particularly evident in all LGM and deglacial samples (Fig. 5). Biosynthesized by certain diatoms in the underside

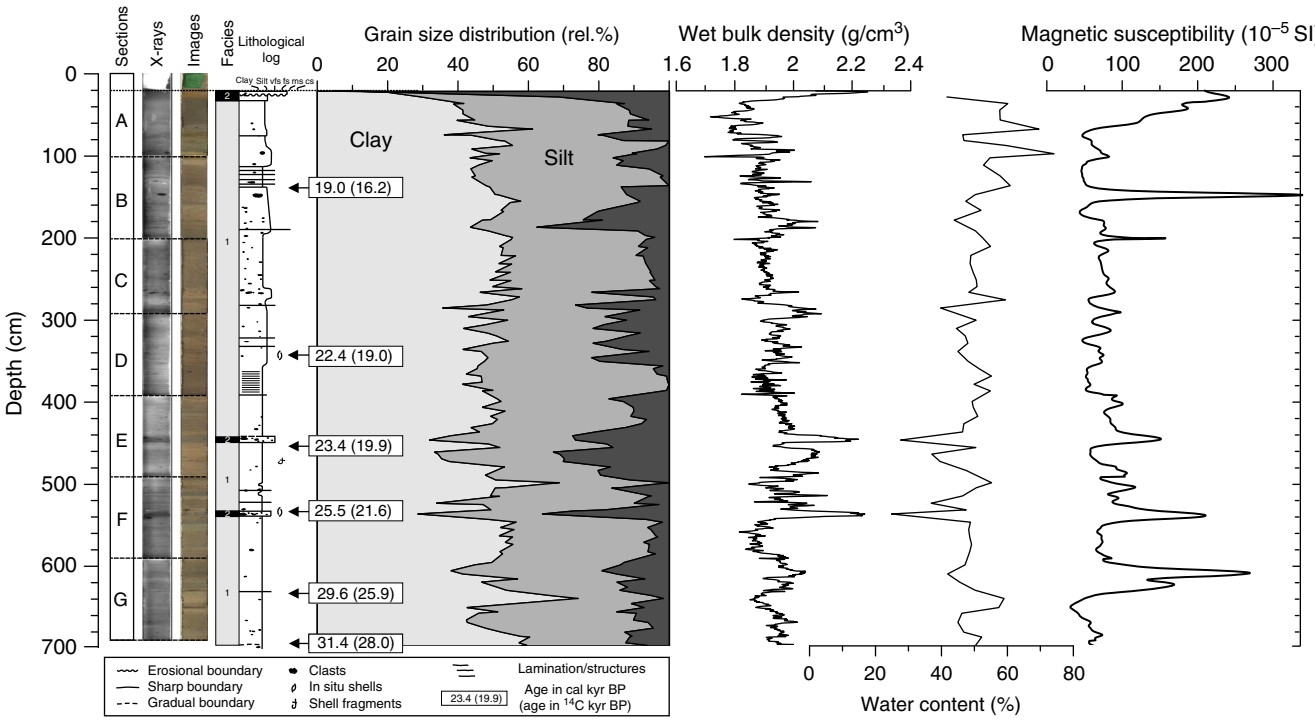

**Fig. 2** Sedimentological and physical properties of GS14-190-01PC (20–700 cm). Available calibrated radiocarbon AMS $^{14}$C datings (uncalibrated ages in parenthesis) are displayed (Table 1)

**Table 1 AMS $^{14}$C datings in cores 33-GC08 and GS14-190-01PC**

| Laboratory code | Core no. | Core depth original (cm) | Core depth converted (cm)[a] | $^{14}$C age (yrs BP) | Error | Carbon source | $\delta^{13}$C (‰) | 2σ max. age (cal. yrs BP) | 2σ min. age (cal. yrs BP) | Median (cal. yrs BP) | Error ± 1s |
|---|---|---|---|---|---|---|---|---|---|---|---|
| UBA-21626 | 33-GC8 | 17.5 | 21.6 | 13007 | ±65 | Mixed benthic/ planktic | −2.5 | 15130 | 14314 | 14800 | 77 |
| UBA-21157 | 33-GC8 | 35 | 41.8 | 13447 | ±49 | *Thyasira* sp. | −9.5 | 15757 | 15273 | 15514 | 64 |
| UBA-21627 | 33-GC8 | 64.5 | 75.9 | 15510 | ±82 | Mixed benthic/ planktic | −2 | 18521 | 18002 | 18272 | 92 |
| UBA-30880 | GS14-190, core B, 37.5 cm | 137.6 | 137.6 | 16183 | ±108 | Planktic forams | −6.1 | 18720 | 19258 | 18966 | 116 |
| UBA21482 | 33-GC8 | 169 | 196.2 | 16516 | ±84 | Mixed benthic/ planktic | −2.2 | 19606 | 19074 | 19360 | 93 |
| UBA-21624 | 33-GC8 | 227 | 268.7 | 17776 | ±94 | Mixed benthic/ planktic | −3.3 | 21193 | 20579 | 20870 | 103 |
| UBA-21625 | 33-GC8 | 244 | 292.5 | 17990 | ±82 | Mixed benthic/ planktic | −3.6 | 21473 | 20868 | 21164 | 92 |
| UBA-21639 | GS14-190, core D, 46 cm | 337.5 | 337.5 | 18950 | ±83 | Mixed benthic/ planktic | −0.4 | 22550 | 22051 | 22358 | 93 |
| UBA-30879 | GS14-190, core E, 63 cm | 454.2 | 454.2 | 19879 | ±161 | Mixed benthic/ planktic | −3.8 | 22935 | 23811 | 23367 | 166 |
| UBA-21638 | GS14-190, core F, 41.5 cm | 532.5 | 532.5 | 21607 | ±86 | Mixed benthic/ planktic | −0.5 | 25714 | 25222 | 25484 | 95 |
| UBA-30878 | GS14-190, core G, 42 cm | 632.6 | 632.6 | 25939 | ±343 | Planktic forams | −1.5 | 28805 | 30518 | 29611 | 345 |
| UBA-30877 | GS14-190, core H, 4 cm | 694.4 | 694.4 | 27959 | ±428 | Mixed benthic/ planktic | −0.1 | 30801 | 32494 | 31407 | 430 |

[a]Indicates converted core depth for GC08 outlined in Table 3

of seasonal Arctic sea ice[35], IP$_{25}$ is commonly used to reconstruct seasonal changes of Arctic sea ice coverage including during the LGM and earlier glacials/interglacials[36,37]. Despite strong bottom current activity in the region during modern times, an allochthonous IP$_{25}$ signal, potentially resulting from lateral sediment transport, can be ruled out since the occurrence of this biomarker in surface sediments from across the Barents Sea (and neighboring regions) reliably reflects the overlying sea ice

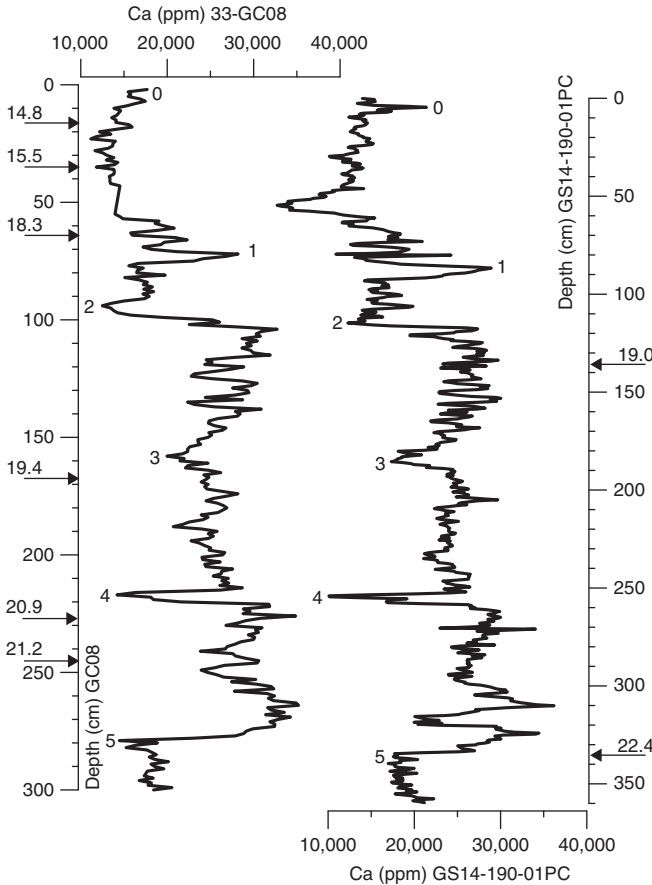

**Fig. 3** X-ray fluorescence records of calcium (Ca) concentrations (ppm) in 33-GC08 and GS14-190-01PC against respective core depths (cm). AMS [14]C (calibrated) datings for each core are shown (see Table 1). Numbers 1–5 indicate correlation tie points (Table 2) between the cores to establish a common depth scale

| Table 3 Age model based on available AMS [14]C dates in cores 33-GC08 and GS14-190-01PC on one common depth scale (GS14-190-1PC) | | | |
|---|---|---|---|
| Age × 1000 (cal yrs BP) | GC08 (depth cm) | GS14-190 (depth cm) | GS14-190 (correlated depth) |
| 14.80 | 17.5 | | 21.6 |
| 15.51 | 35 | | 41.8 |
| 18.27 | 64.5 | | 75.9 |
| 18.97 | | 137.6 | 137.6 |
| 19.36 | 169 | | 196.2 |
| 20.87 | 227 | | 268.7 |
| 21.16 | 244 | | 292.5 |
| 22.36 | | 337.5 | 337.5 |
| 23.37 | | 454.2 | 454.2 |
| 25.48 | | 532.5 | 532.5 |
| 29.61 | | 632.6 | 632.6 |
| 31.41 | | 694.4 | 694.4 |

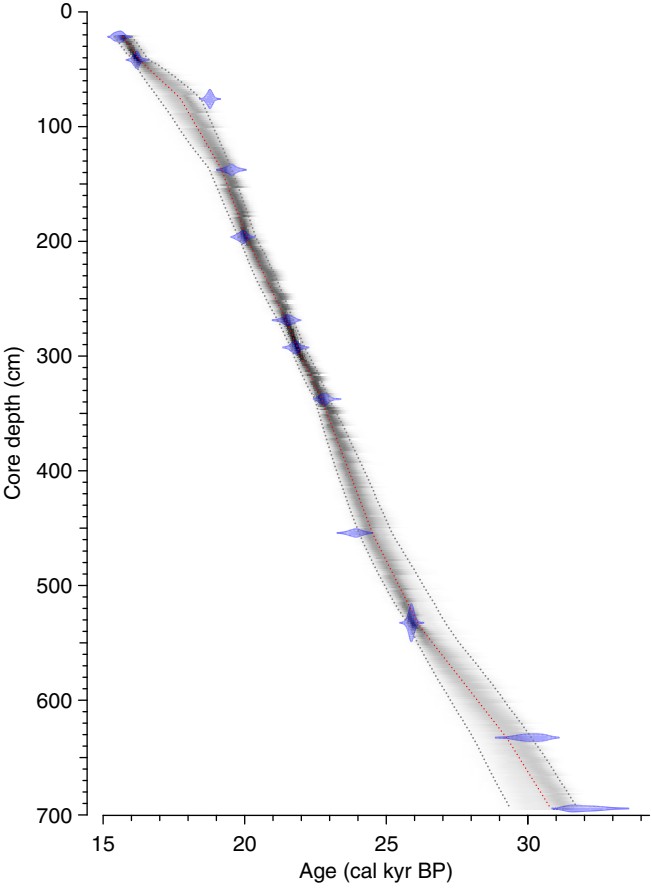

**Fig. 4** Age model. Age-depth relationship of the upper 700 cm of sediment core GS14-190-01PC using a Bayesian age modeling (Bacon v2.2) approach[34] on 12 AMS [14]C dates

| Table 2 Different depths of correlation tie points 1–5 in cores 33-GC08 and GS14-190-01PC based on XRF-Ca records | | |
|---|---|---|
| Tie point number | Core GS14-190 depth (cmbsf) | Core GC08 depth (cmbsf) |
| 0 | 0.1 | 2 |
| 1 | 87.1 | 72.2 |
| 2 | 115.2 | 94.8 |
| 3 | 185.9 | 160.0 |
| 4 | 253.8 | 216.0 |
| 5 | 333.9 | 278.7 |

conditions. Thus, absent IP$_{25}$ has been reported for sites of year-round ice free conditions (including those close to the GS14-190 core site). Relatively low concentrations are found close to the winter sea ice margin, with highest abundances in regions of more extensive seasonal sea ice cover, which are mainly >76°N[38] (Supplementary Figure 1). As such, vertical transport dominates the IP$_{25}$ sedimentary signal across the study region. In GS14-190, highest IP$_{25}$ concentrations occur during the LGM, with lower values before and after. The open marine biomarker dinosterol mainly follows the variability in the IP$_{25}$ record during the LGM (Fig. 5), while the number of planktic foraminifera is also highest during the LGM, consistent with the widely observed elevated biogenic calcite content in LGM sediments along the entire

Barents Sea continental margin[21,39–41]. Both IP$_{25}$ concentration and planktic foraminifera drop to minimal values during the initial deglaciation (Fig. 5). A gradual increase in dinosterol and planktic foraminifera occurs towards the final deglacial phase. IRD supply is reflected by the coarse ( > 250 μm) fraction and the high-resolution Zr/Al record. Higher proportions of IRD are observed during the middle to late LGM and the initial deglaciation. The most prominent IRD pulse is centered at ~19.0 ka BP (Fig. 5). All data are available in Supplementary Table 2.

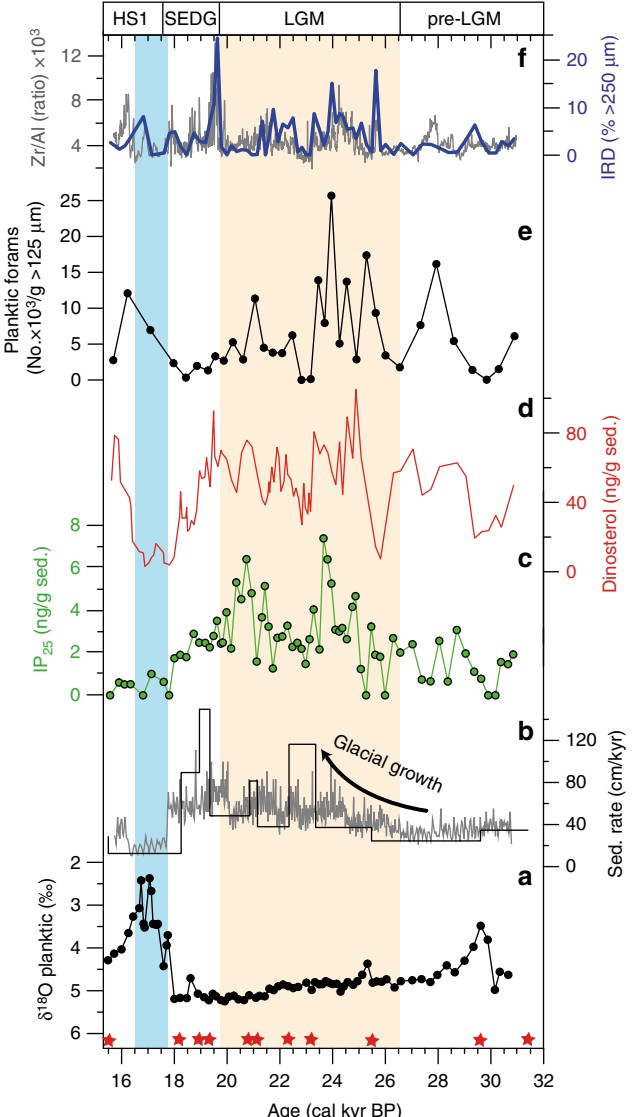

**Fig. 5** Proxy data of sediment cores GC14-190-01PC for the late glacial period (ca. 31–15 cal. kyr BP). **a** Planktic $\delta^{18}$O (‰) measured on *Neogloboquadrina pachyderma* sin. **b** Sedimentation rates (cm/kyr) inferred from Bayesian age-depth modeling (Bacon v2.2) (black line) and linear interpolation (gray line) between 11 AMS$^{14}$C dates (red asterisks at bottom). **c** IP$_{25}$ concentration (ng/g Sed) (green dots). **d** Dinosterol concentration (ng/g Sed). **e** Number of planktic foraminifera in coarse fraction (>125 μm) in thousands. **f** Ice rafted debris (IRD) inferred from % of >250 μm coarse fraction (blue line) and XRF based Zr/Al ratio (gray line). HS1: Heinrich Stadial 1, SEDG: Shelf Edge Deglaciation, LGM: Last Glacial Maximum (beige colored)[19]. The bluish rectangle marks the freshwater release during Svalbard Barents Sea ice sheet collapse

## Discussion

Full glaciation of the SW Barents Sea shelf during the LGM is illustrated by the presence of glacigenic debris flows along the continental margin[24]. The synchronized age of these mass transport deposits with the occurrence of IRD layers in dated deep sea sediments reveal a western shelf edge position of the ice sheet at ~24–23 ka BP[20,42]. Inception and progressive advance, however, remain poorly constrained and built largely upon the geomorphological footprint of the advancing ice sheet towards the shelf break[26]. Our new data from the SW Barents Sea margin suggest a smooth pattern of gradual ice build-up. A fourfold

increase in sedimentation rates (20–80 cm kyr$^{-1}$) from glacial inception at ~30 ka BP to the LGM (~21 ka BP), superimposed by short events of maximum rates during first shelf edge glaciation (24–23 ka BP) and initial ice sheet collapse (~20–19 ka BP) indicate a slow, but continuous movement of the ice sheet towards the shelf break (Fig. 5). The latter is evident by enhanced IRD supply at the start of the LGM (~25 ka BP), suggesting increased iceberg calving rates close to the shelf edge (Fig. 5). The positioning of moderately abundant IP$_{25}$ concentrations in pre-LGM deposits indicate the presence of the polar front at the study area and, therefore, subduction of Atlantic water-derived water masses below seasonally sea ice covered surface waters. The accompanying high abundance of calcareous organisms[21,39] would have resulted from the prevalence of sub-surface Atlantic water masses penetrating the eastern Nordic Seas below seasonal to perennial sea ice cover[14,43]. In contrast, further north, and towards the Fram Strait, sea ice cover was more permanent and primary productivity significantly reduced between 28 and 26 ka BP[44].

With the grounded ice sheet approaching the shelf edge, as shown by the steadily increasing sedimentation rates and frequent IRD delivery, IP$_{25}$ and dinosterol concentrations both increase significantly (Fig. 5), however, with some excursions to lower values paralleled by a decline in planktic foraminifera (Fig. 5). The high abundance of IP$_{25}$, dinosterol and calcareous organisms indicates significant primary productivity along an active sea ice margin. The latter requires seasonal open water conditions, which we argue constrains the presence of a grounded ice sheet at the shelf edge to maintain formation of coastal polynyas. Short-term phases with lower sea ice diatom blooms and reduced phytoplankton growth reflect more extensive sea ice cover. Our proposal of widespread polynya occurrence in front of the SBIS is supported further by high IP$_{25}$ concentrations in marine biomarker- and calcareous-rich deposits along the western Svalbard margin (Fig. 6)[44]. Along the entire western and northern Svalbard-Barents Sea margin (Fig. 7), opposing effects of katabatic winds—blowing sea ice westwards, away from the ice sheet—and an expanded polar front in the Nordic Seas—pushing sea ice limits eastwards towards the ice sheets—created a marginal ice zone that allowed for high production of sea ice diatoms, marine phyto- and zooplankton, and (limited) deep water formation. Coupled to this, polynya formation would also have been sustained by the aforementioned ongoing upwelling of inflowing warm sub-surface Atlantic water. The presence of a widespread sea ice-free corridor in front of the SBIS during the LGM was spatially constrained, however, with near perennial sea ice cover further west, as evident from poorly ventilated deep waters and decreased benthic $\delta^{13}$C values in the Nordic Seas (Fig. 6)[9,14]. This scenario contrasts earlier inferences of pulse-like heat transport by inflowing Atlantic water that controls eastern Nordic Seas sea ice dynamics[21,44]

In a new Eurasian ice sheet model for the last deglaciation, Patton, et al.[45] revealed the complexity of the asynchronous deglaciation pattern of the marine-based SBIS and its sensitivity to climate and oceanic forcing, as well as internal ice dynamics. However, uncertainties remain as the empirical records used by the model are fragmentary and poorly constrained, chronologically. To date, the most robust timing for the initial destabilization of the SBIS is established for the western Svalbard margin, with age control of IRD pulses and the onset of hemipelagic sedimentation on the shelf at ~20 ka BP[46–48]. In the present study, we confirm this timing of regional ice sheet disintegration for the SW Barents Sea margin. Distinct pulses of IRD and highest linear sedimentation rates (~140 cm ka$^{-1}$) occur between ~20–19 ka BP and suggest enhanced iceberg calving and debris release from the retreating grounding line (Fig. 5). This

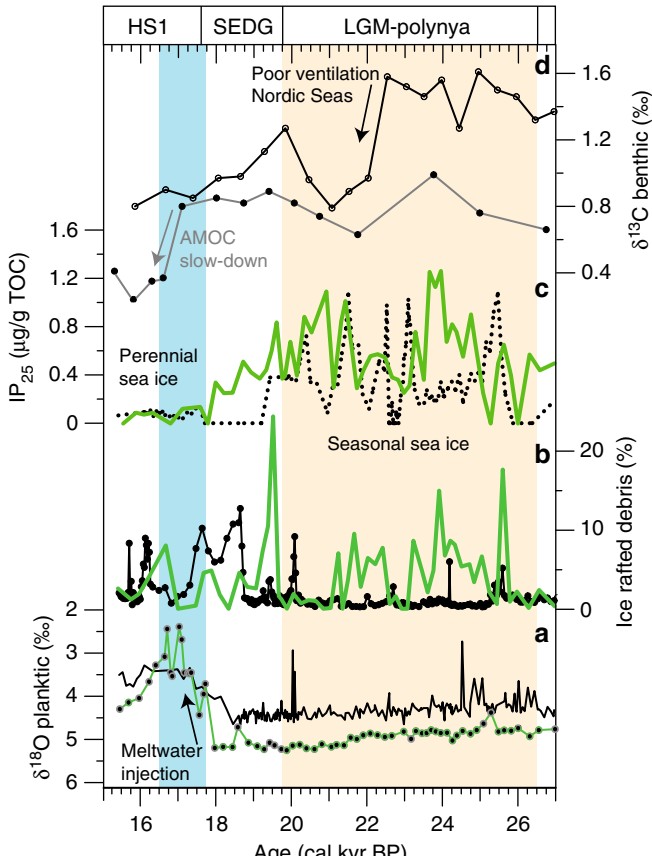

**Fig. 6** Sea ice and ice sheet dynamics along the western/northern Barents Sea margin during the LGM. **a** Planktic δ[18]O (in ‰) and **b** ice-rafted debris (% >250 µm) records from the southwestern (this study: green) and northern (PS2138-1: black) Barents Sea margin[52]. **c** IP[25] in sediments off northwestern Svalbard (MSM05/5-712-2) (dotted line)[44] and southwestern Barents Sea (green line) (this study). **d** Benthic foraminifera δ[13]C in the Norwegian Sea (PS1243; black)[14] and the North Atlantic (ODP Site 980; gray)[59]. Timing of ventilation changes in Nordic Seas and AMOC slow down is marked by arrows. See references for exact location of these cores. Legend of colored rectangles as in Fig. 5

observation agrees with previous inferences of an open corridor between the SBIS and Fennoscandian ice sheets at ~18.6 ka BP[49]. This initial instability is likely induced by increasing northern summer insolation[19] – a fact that has also been advocated for the ice sheet disintegration on the Antarctic Peninsula and East Antarctica in the southeastern Weddell Sea[50].

The commencement of ice sheet disintegration in the SW Barents Sea is not, however, followed by large-scale meltwater injections into the Norwegian Sea. Indeed, heavy δ[18]O (~5 ‰), slightly reduced IP[25] and dinosterol concentration, as well as moderately abundant planktic foraminifera do not support proximal water mass perturbations (Fig. 5). Although a conspicuous δ[18]O depletion around 20–19 ka BP (Fig. 6) may be correlated to other records in the Nordic Seas[51], minor meltwater injections had no impact on polynya stability. In contrast, further collapse of the northern SBIS, as documented by massive IRD pulses starting ~19 ka BP (Fig. 6)[52], changed the physical conditions for perennial polynya activity in the northern region. Massive iceberg and freshwater export through the Fram Strait (Fig. 6) and sub-surface cooling west of Svalbard[41] facilitated permanent sea ice cover between ~19 and 17.6 ka BP[44], preventing any polynya activity, as confirmed by the absence of IP[25] (Fig. 6)[44] and reduced phyto- and zooplankton productivity

indicators[41]. In the SW Barents Sea, decreased ventilation of Atlantic intermediate waters and progressive surface cooling in the study area[39] between 18.3 and 17.7 ka BP may have increased the sea ice concentration, as indicated by a gradual decline in IP[25] (Fig. 6). However, this climate deterioration was not severe enough to prevent polynya activity entirely. A likely explanation for the variability in sea ice conditions along the western Svalbard/Barents Sea margin could be the Atlantic water current-induced northward transport of iceberg armadas released by the southwestern SBIS in concert with increased iceberg export from the retreating northern SBIS southwards via Transpolar Drift through the Fram Strait. High densities of icebergs west of Svalbard likely facilitated sea ice growth as a result of lower sea-surface temperatures induced by latent heat of melting[53], which ultimately would have resulted in long-lasting permanent sea ice coverage in the Fram Strait[44,54]. Nonetheless, intense IRD deposition due to massive iceberg calving along the SBIS margin between 20 and 17.5 ka confirms the significant contributions of IRD-rich sediments derived from the SBIS to the IRD belt in the North Atlantic prior to the Heinrich Stadial 1 (HS1; ~17.5 ka)[55,56].

Conditions changed dramatically when the marine-based SBIS collapsed. With gradually increasing summer insolation, and increases in atmospheric temperature and sea level[19], the ice sheet disintegrated rapidly after ~17.6 ka, with a modeled rate of ~670 gigatonnes per year (Gt a$^{-1}$)[45]. This enhanced rate is reflected in a large δ[18]O planktic anomaly in the study area[57] and indicates a massive meltwater injection into the eastern Nordic Seas[5] and Arctic Ocean[22] during the onset of HS1 (Fig. 6). The southwestern SBIS has often been discussed as a major source of freshwater perturbations in the Nordic Seas during HS1[5,51,57,58]; however, until now, there has been a paucity of direct indication for such changes due to the prevalence of mass flow deposits along the margin. The data presented herein, however, constitute further evidence of massive meltwater injections in the eastern Nordic Seas likely being sourced from the collapsing southwestern SBIS. In addition, freshwater outburst from ice-dammed lakes in NW Russia associated with the SBIS collapse may have contributed to the freshwater δ[18]O anomaly as well, which caused significant perturbations of the AMOC during the last deglaciation[59]. Indeed, the response of the sea ice coverage to this freshwater outburst was immediate, with consistently low IP[25] and dinosterol concentrations indicative of near-perennial sea ice cover and an inactive coastal polynya in front of the retreating SBIS (Fig. 5). IP[25] concentration also remained consistently low along the entire western SBIS margin during HS1 (Fig. 6), implying mainly severe sea ice conditions relative to the LGM, which likely extended to the southern end of the Norwegian Sea[60], and a significant weakening of the AMOC during HS1[9,59,61] illustrated by a considerable decrease in North Atlantic benthic δ[13]C values (Fig. 6). Due to the perennial sea ice coverage throughout the Nordic Seas during HS1, basin-wide accumulation of heat – likely a result of subdued warm Atlantic water inflow[62], and/or isolation of the deep Nordic Seas[9] – was probably a crucial pre-requisite for the recovery of the climate system, as expressed by the gradual rise in phyto- and zooplankton in surface waters at the end of HS1 (Figs. 5 and 6).

From a broader perspective, the finding of large scale polynyas in front of LGM ice sheets in this study support recent inferences by Keigwin and Swift[8] that formation of glacial bottom waters in the western North Atlantic may be derived from sea ice controlled brine rejections within polynyas facilitated by katabatic winds blowing off the Laurentide Ice Sheet. Moreover, the prevalence of large-scale polynyas in front of these growing ice sheets during the LGM presented an ideal moisture supply to sustain such growth. This information is crucial for validating numerical

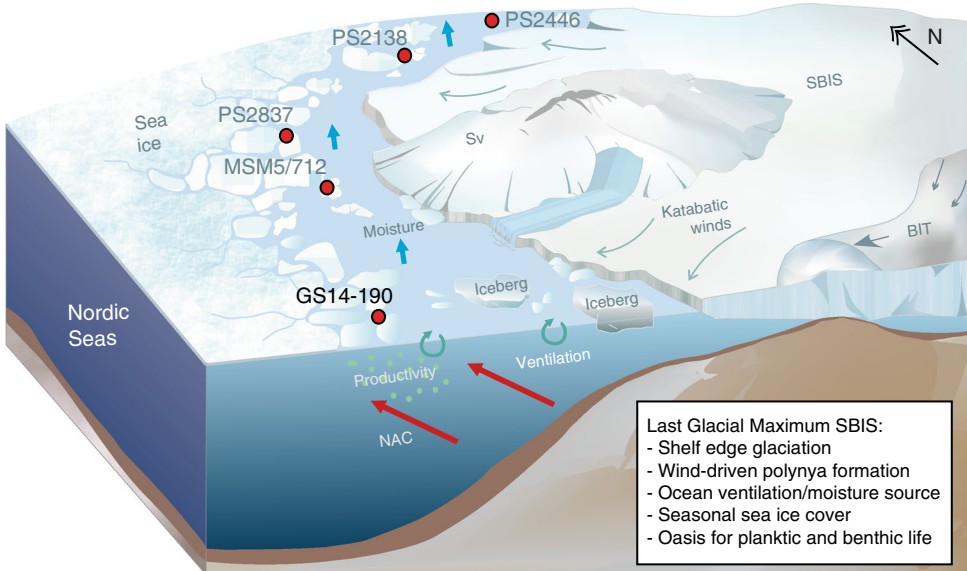

**Fig. 7** Schematic illustration of the polynyal activity in front of the western and northern Svalbard-Barents Sea ice sheet during the LGM as reconstructed from 5 sediment proxy records (Supplementary Table 1). Polynya activity is constrained by relatively high sea-ice diatom ($IP_{25}$), marine organic phytoplankton and calcareous zooplankton production in all displayed sediment cores (this study)[16,44,54,69,70] supported by sub-surface/intermediate inflow of Atlantic-water-derived waters (NAC) and katabatic winds. Coastal polynyas along the entire Svalbard-Barents Sea margin provided a constant source of moisture that sustained build-up of glacial ice, ventilation of deeper waters in the glacial Nordic Seas, and remained a refuge for marine and higher trophic terrestrial life in a polar desert. Sv: Svalbard, SBIS: Svalbard-Barents Sea Ice Sheet, BIT: Bear Island Trough, NAC: North Atlantic Current

models on ice-sheet configuration[20] and provides an important baseline to test ice sheet dynamics during past glacial/interglacial cycles[63]. Furthermore, the possibility that wind-driven polynyas in front of large-scale marine-based ice sheets are potential refuges for higher-trophic life in the Arctic during glacial times requires further attention, particularly after recent findings in Antarctica (i.e., the Ross Sea) where marine and terrestrial faunal changes were found to be highly correlated with sea ice dynamics in coastal polynyas[12]. It has also been postulated that the existence of polynyas during glacial periods was essential for the survival of marine and terrestrial life[17]. Indeed, previous reports of unusually high abundances of the benthic foraminifer *Cassidulina neoteretis* and smaller-sized planktonic specimens of *Turborotalita quinqueloba* in LGM sediments off western Svalbard[14,40,48,51] indicate the existence of a productive benthic oasis in an otherwise glacial desert at this time, with nutrient supply and organic matter production in a marginal ice zone supported by upwelling of nutrient-rich Atlantic-derived water masses.

Hence, we propose the presence of large-scale coastal polynyas in front of the extensive NW Eurasian ice sheets during the Last Glacial Maximum in an otherwise Arctic desert, characterized by a perennial sea-ice cover. This conclusion is built on the co-existence of high amounts of sea-ice algae, primary and secondary plankton producers, and a high diversity of benthic and planktic fauna[14,40,48,51] in an environment that is characterized by a marginal sea ice zone in close proximity to the Svalbard-Barents Sea continental margin and upwelling of nutrient-rich, sub-surface flowing Atlantic waters during this period. Polynya activity likely collapsed when ice sheets disintegrated and freshwater injections to the eastern Nordic Seas caused rapid sea ice expansion in the entire Nordic Seas and AMOC weakening during Heinrich Stadial 1, reflected by a significant $\delta^{18}O$ anomaly and near complete cessation of sympagic and pelagic productivity. Overall, our findings confirm the crucial importance of coastal polynyas for survival of planktic and benthic species during otherwise harsh glacial conditions in the Arctic.

## Methods

**Sedimentological properties**. Whole core measurements, i.e. wet bulk density (WBD) and magnetic susceptibility (MS) were conducted on the gravity and piston cores using the Standard MSCL-S core logger (GeoTek Ltd., UK) at 1 cm resolution with 5 s measurement time. MS measurements for whole cores were done with a Bartington MS2C loop sensor with 130 mm coil diameter. For control of the MS2C sensor, a certified sample piece with known magnetic susceptibility was measured. After lengthwise splitting and surface cleaning core surface images were taken with the GeoScan IV color line-scan camera. The camera was equipped with a Nikon AF Nikkor 50 mm f/1.8D lens and three detectors using three 2048 pixel charge-coupled device CCD arrays for red, green and blue light. The core surface was continuously imaged with 100 μm down and cross core resolution. Each core section was measured with an X-rite ColorChecker as reference for basic color control.

X-ray images (XRI) of split cores were taken with the Geotek MSCL-XCT (Geotek Ltd., UK). The device is equipped with a Thermo Kevex PSX10-65W X-ray source (Thermo Fisher Scientific Inc., USA) and a Varian PAXScan 2520 V X-ray detector (Varian Medical Systems, Inc., USA). Voltages of 87 and 120 kV and electric currents of 125 and 140 μA were used for imaging of both cores.

The grain size distribution (0.4 μm – 250 μm) was determined with a Coulter LS 200. To prevent charging and agglomeration of particles, de-carbonated samples were treated with 5% sodium pyrophosphate ($Na_4P_2O_7 \times 10H_2O$, MerckPA) and sonicated. To compensate for elevated silt contents compared to traditional methods (Pipette and Sedigraph), we applied the method published by Rise and Brendryen[64]. Grain sizes >250 μm were determined by dry sieving on a duplicate of each sample (relative error ± 10%) and identified as ice-rafted debris (IRD).

**Inorganic and organic geochemistry**. X-Ray fluorescence (XRF) core logging was carried out with the Standard MSCL (MSCL-S) core logger (GeoTek Ltd., UK) and an attached DELTA Handheld XRF sensor. The XRF sensor is equipped with a 4-W Rh Tube anode and Si drift detector. Prior to core measurements the XRF sensor was standardized and SRM 2710a Montana soil I standard sample[65] was stationary measured for sensor-control purposes. Down core XRF measurements were taken incrementally along the longest axis in the centre of the split core surfaces with 0.5 cm steps. Two measurements in succession with 40 keV and 10 keV currents and 10 s exposure time each provided spectra covering chemical elements from Mg to Pb, of which only the zirconium (Zr), aluminum (Al), and calcium (Ca) concentrations (ppm) were used for this study.

Analyses of total (TC) and organic carbon ($C_{org}$) were performed with a LECO SC-632. For TC determination subsamples of 300–400 mg were combusted at 1350 °C and the release of $CO_2$ determined. For $C_{org}$ analysis, sub-samples of 400–450 mg were placed in carbon-free pervious ceramic combustion boats. These were placed on a heating plate at 50 °C ( ± 5 °C) and treated with 10 vol.% hydrochloric acid (HCl) to remove inorganic carbon (carbonate) and subsequently rinsed with distilled water and dried in the drying oven prior to analysis. Results

are given in weight percentage (wt. %) and the standard deviation of the TC and $C_{org}$ measurements based on the repeated measurement of a standard was ± 0.026 w. t% ($1\sigma$, $n = 8$) and ± 0.028 wt. % ($1\sigma$, $n = 11$), respectively.

The sea ice biomarker $IP_{25}$[66] was quantified following addition of an internal standard (9-octylheptadec-8-ene, 0.1 µg) to freeze-dried sediments (ca. 1–3 g), extraction (2:1 v/v dichloromethane:methanol; $3 \times 2$ mL) and purification of extracts using silica column chromatography (hexane, 6 mL). Further purification to remove saturated hydrocarbons was achieved using Ag-ion chromatography (Supelco Discovery® Ag-Ion; ca. 0.12 g; 1 mL of hexane), followed by separate elution of unsaturated hydrocarbons including $IP_{25}$ using acetone (2 mL). Sterol fractions were obtained following internal standard addition (5α-androstan-3β-ol, 10 µL; 10 µg mL$^{-1}$) to freeze-dried sediments, saponification (2 mL of 5% m/v KOH in 9:1 methanol:milliQ water; 60 min; 70 °C) and back extraction into hexane ($3 \times 2$ mL). Silica column chromatography was used to remove impurities, including hydrocarbons and long-chain ketones (7:3 DCM:hexane; 6 mL), and for sterol elution (4:1 hexane:methyl acetate; 6 mL). Partially purified $IP_{25}$- and sterol-containing fractions were analyzed using gas chromatography-mass spectrometry (GC-MS) according to established methods[67]. $IP_{25}$ and sterols were identified by comparison of total ion current (TIC) mass spectra and retention indices to those of authentic standards or those presented previously[68]. Biomarker quantification was carried out in Selective Ion Monitoring (SIM) mode by comparing the peak areas of molecular ions of individual biomarkers to those of the internal standards, followed by correction for sediment mass and ion fragmentation efficiency. In the current study, we confine the sterol analysis to that of dinosterol.

For planktic foraminiferal concentration/abundance the sediment core was sampled in ca. 20 cm intervals. Approximately 300 specimens from >125 µm fraction were picked and identified per sample, using a micro-splitter if the number of foraminifera was higher. Only one sample contains less than 200 planktic foraminifera (depth 368 cm); one sample (depth 345 cm) is barren of foraminifera. For oxygen stable isotope analyses ($\delta^{18}O$), monospecific planktic foraminifera *Neogloboquadrina pachyderma* (sinistral (sin.)) were hand-picked from the >100 µm size fraction. For each interval, 5–10 specimens were analyzed. $\delta^{18}O$ values were determined using a Thermo Scientific MAT253 mass spectrometer coupled to a Finnigan Gasbench II at UiT, The Arctic University of Norway in Tromsø. Results are reported relative to the Vienna Standard Mean Ocean Water (VSMOW) standards in per mil (‰) notation. External precision was ± 0.07‰ based on analysis of NBS-19.

## Data availability
All data are given in Supplementary Table 2.

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

## Acknowledgements

This research was supported by MAREANO (www.mareano.no), the Norwegian Deep-water Program (NDP) and the Norwegian Research Council (RCN grants 223259, 255150). We thank the captain and crew on RV *G.O. Sars* for cooperation during data acquisition and Irene Lundquist for the artwork.

## Author contributions

The main idea was developed by J.K., and J.K., S.T.B. wrote most of the text. The IP$_{25}$ and dinosterol analyses were made by D.K., while stable isotope work and faunal counts were performed by G.P. and P.J. Lithological description and non-destructive XRF analyses were provided by N.B. and M.K. Geological and geophysical data acquisition was co-ordinated by R.B., L.R. and V.B. All authors discussed the results and commented on the manuscript.

## Additional information

**Competing interests:** The authors declare no competing interests.

