## [Peer Review File · Nature Communications]

Reviewers' comments:

Reviewer #1 (Remarks to the Author):

This study provides important evidence for the presence of (seasonal) polynyas in front of the Svalbard-Barents Ice Sheet during the last glacial maximum. Although this main finding is clear and well supported by the provided data, there are few issues I list below that should be addressed before publication.

The title should be adjusted so it better describes the actual work presented in this paper. This study presents evidence for the presence of polynyas in front of the SBIS during the LGM. The current title is too broad and vague.

The chronology needs to be improved. Fitting a third-order polynomial function through the dates is clearly not ideal. Some radiocarbon date probability distributions fall entirely outside the final age-depth curve (e.g. UBA-21627, UBA-21482, UBA-21639). A Bayesian approach to the age modelling would be preferred (using Oxcal or Bacon), but even a linear interpolation between the median calibrated ages would give a more reliable result than the one presented.

The modelled sedimentation rate is then used to argue for a slow and continuous advance of the ice sheet towards the shelf break (lines 183-184, lines 194-195). There is less than 1 date per 1000 years and the fitted curve is unrealistically smoothed. Based on Figure 4, the highest sedimentation rates seem to actually occur just after 20 ka, but this is where 2 dates are falling of the age model curve. This later period of maximum sed rate would also match nicely with the presented IRD and Zr/Al data, as written in lines 235-236. Sometimes the modelled age-depth curve is used, sometimes the linear interpolation.

The two studied cores are matched up using XRF Ca records and radiocarbon dates are subsequently moved from one depth scale to the other. This method should be described in more detail. In Table 2, please show both the original and converted core depths. The detailed description together with Figure 3 might be better placed in supplementary information.

The IRD pulse at 19 ka shows a jump from 0-5% to ca. 25%. All other measured variables seem unaffected by this change. If suddenly a quarter of the material is derived from ice-rafting, this should be visible as a slight drop in concentrations of forams, dinosterol, IP25 per gram. How come this doesn't affect any other proxies?

Figures 2, 5, and 6 all show data from the studied sediment core, but they are not all showing the same interval. I suggest to add a depth scale to figures 5 and 6 besides the age – axis. In the top of Figure 2, the sand content increases dramatically, but this is not shown in figure 5 or 6? Figures 5 and 6 show nearly the same time interval but not entirely, I suggest the authors correct this.

The comparison of data in Figure 6 forms the basis for the interpretation of large-scaly polynyas in front of the SBIS. It is not clear from the figure and the caption which curve belongs to which core (e.g. the dotted line of IP25). Curve (a) shows oxygen isotopes from two cores, with about 1 ka difference in timing of the meltwater pulses, but this is not discussed. The same time lag can be seen in the IP25 records. Curve (b) doesn't show a comparison of IRD between two records but rather uses Zr/Al, why? Curves (c) and (d) in Figure 6 seem to only be included to support a very short statement in lines 283-284. This should be expanded. Neither of these sites are shown on a map.

Several of the cores in Figure 7 are not mentioned in the text.

Line 74 and 314: correct oases to oasis.

Figure 1A: In the legend, the bathymetry color scale is a continuous gradient of shades of blue. The map itself however doesn't follow this color scheme: the deeper sites are much more purple.

Figure 1A: Add label for the Bear Island Trough

Line 108: Core name ends in -01PC on the map, but -1PC in the text. Use a consistent name.

Line 109: Water depth of core site: is it 990m (main text) or 949m (Table 1)?

Lines 144-145: remove "that operates with a standard =0)". This is not entirely correct. The difference between the atmospheric and marine curve is approx. 400 years on average but this is not the same as R . Simply state that $\Delta R = 0$ years is enough.

Figure 5 and 6: Describe the red and blue shading in the plot. Red is LGM? What is blue?

Figure 5b: Sedimentation rates based on the age model of Figure 4. Why are there so much high frequency wiggles on this curve? The age model is a smooth curve so why would there be so much small variability in the sedimentation rate?

Figure 5e: The left of this curve seems cut off. Is there a data point before 16 ka which is not shown?

Reviewer #2 (Remarks to the Author):

Knies et al present new IP25 data from off the Barents Sea and use this to show seasonal sea-ice conditions during the glacial period, which is linked to polynya formation by katabatic winds of the SBIS. This work provides further support for the concept of a seasonal ice free corridor along the Norwegian coast and Barents Sea edge – a concept that has been proposed eg since Hebbeln et al 1994 Nature, and the several other papers cited by the authors. The advance here is that they provide sea-ice biomarker data from an additional site (building on work from the Fram St), that complements earlier work based largely on foraminiferal abundance and stable isotopes. The study provides a valuable record that helps with the development of a strong body of evidence for seasonally ice free conditions in this region. The main interpretations are sensible and follow from the data (and earlier studies) and this is a useful addition to the literature, although arguably not providing any significantly new advance in concepts or ideas. I have only minor suggestions for revisions, largely urging some caution in places and flagging up to readers where there are mitigating controls, although the robustness of the IP25 record for recording in situ conditions does warrant serious attention.

My main concerns regards the lack of consideration of transport of the material containing the biomarkers. These are predominantly in the fine fraction which is highly mobile and can be transported by currents 100s to 1000s of km (eg McCave et al 2001, Science and refs therein; Filippova et al 2016 Paleocyanography), with the finer clay fraction being transported across large basins (eg Fagel and Mattielli 2011 Paleocyanography). The study site here is one that (as stated by the authors) is strongly influenced by the northward transport of the Norwegian Atlantic Current, which has been shown to be capable of sorting and transporting coarse silts (Tegzes et al 2017, The Holocene) and building sediment drifts. How do the authors of this study constrain the provenance of the sediment size fraction containing the biomarkers and how do they rule out that it is not an allochthonous signal? Their record could simply be one of advected biomarkers from further south. Complementary data from sand size proxies such as foraminifera helps provide support for their interpretation of in situ seasonal sea-ice cover, but the authors must do a lot more to inform the reader of this major weakness in their IP25 evidence, and it would have been useful if they presented a case to rule out advection artefacts on their IP25 record.

I am supportive of the work, but am not convinced about how this represents a major advance from what the community already knew, as shown by the useful references to earlier work provided, which reach similar conclusions. Please could the authors highlight specifically how this is an advance from earlier work – more than just listing earlier evidence as being speculative whereas theirs is definitive. Thanks.

The assumptions made in the age model construction are likely flawed – specifically the use of a 400 yr reservoir age. This is likely to be variable and 400yr seems far too low. Mixed benthic/planktics were used (including *N pachyderma*?), yet studies using these signal-carriers from Nordic Seas and N Atlantic show reservoir ages of up to 2000 years (Voelker et al 1998, Waelbroeck et al 2001, Sarnthein et al 2001, Stern and Lisiecki 2013, Thornalley et al 2011 and 2015, Ezat et al 2017). Greater consideration of variable reservoir ages should be made, including error estimates and at least mentioning the possibility of much higher reservoir ages in the text, and briefly saying how this would shift the timing of events to younger calendar ages.

The study only uses concentration per gram of sediment, which can be susceptible to changes in the basin wide input of terrigenous material. Usually the effects of changes in sediment flux and focussing are constrained using Th-normalization to better understand sediment fluxes. It does not seem like the major results would be countered by this effect (because IP25 conc increases at a time when sed rate also increases; and fortunately the authors have not then taken the next step of converted to a flux per year which would have been erroneous without Th-normalization) , however it significantly limits the interpretation of downcore trends other than the major increase and decrease, since transient decreases in concentration may be due to episodic basin input of terrigenous material.

L55 (and later text) – I would be careful overstating the role of brines in AABW formation – important, but perhaps it is useful to mention/reference the other processes, citing eg Toggweiler and Samuels 1995 (J Phys. Oceanogr., 25, 1980)

L276 – please reword to avoid phrases like “leaves little doubt that”. It sounds like you are trying too hard to convince the reader by saying these phrases, rather than simply providing the evidence to support your case. Moreover, previous studies (eg Dokken and Jansen 1999, Nature) have provided similar evidence for $\delta^{18}O$ anomalies that are as large as the planktic $\delta^{18}O$ presented here, so it is unclear why this is such a major advance and now leaves us in no doubt (if we were before). I would instead just say that the large planktic $\delta^{18}O$ provides further support for the role of SBIS in meltwater input at this time.

L320 – please provide evidence or a reference for the high diversity of planktic, and especially benthic, diversity...none seems to be provided.

Fig 7 – would be improved if the various evidence at each sites was somehow shown or listed in the caption or elsewhere. As a reader it was not clear what core each site showed, without doing detailed searching of the text and linking up shorts comments in the text with the diagram – nicer if the authors can help the reader here.

Reviewer #3 (Remarks to the Author):

The study by Knies et al. deals with an important subject in marine glacial climate reconstructions, namely the crucial role of polynya systems as drivers on both marine biotic production and ocean circulation in a glaciated polar region. Overall I quite like the approach in general. But there are some shortcomings, which, I believe, can be dealt with by the authors during revision. One is to do with the fact wind forcing has on not only creating open water leads (polynyas) and enhancing

local bioproductivity, but also causing upwelling that would inevitably lead to increased ocean circulation such as advection of Atlantic waters from subsurface-intermediate depths with shallow overturning at probably also an intermediate water depths. Such situations have been suggested/interpreted before by various authors in studies from continental arctic margins and open polar ocean (Noegaard-Pedersen et al, Taldenkova et al., Bauch et al...). Although the authors mention it I believe this topic deserves some better strengthening and further elaborating... especially because it is still consensus among many of the "paleocommunity" that ocean circulation in the polar ocean was very sluggish and insignificant.

I support the idea that the onset of collapse of the Barents ice caps and shelves caused a dramatic change in the habitat of marine plankton biota, at least in the proximity of thick glacier ice. To what areal extent, however, remains rather speculative as the center of productions could have easily shifted elsewhere like into the central parts of the Nordic seas (e.g., Iceland Sea) where the meltwater lid was very likely of lesser thickness.

And also, while the Barents Sea ice sheet was quickly gone during early deglaciation, I wonder if there is more supporting evidence (IP25 and other proxies) of a continuation of the polynya processes described by the authors until 15ka in other areas where the ice sheet (and katabatic winds) remained close by in the hinterland for a substantial longer time during deglaciation. The region west of Spitsbergen with its narrow shelf is an adequate area to look into...and the cores are listed here.

Revisions recommended by reviewer 1

This study provides important evidence for the presence of (seasonal) polynyas in front of the Svalbard-Barents Ice Sheet during the last glacial maximum. Although this main finding is clear and well supported by the provided data, there are few issues I list below that should be addressed before publication.

Reply: We thank the reviewer for the positive feedback!

The title should be adjusted so it better describes the actual work presented in this paper. This study presents evidence for the presence of polynyas in front of the SBIS during the LGM. The current title is too broad and vague.

Reply: We adjusted the title of the manuscript as requested. Still, we wish to reach out to a broader readership of Nature Communications interested in Arctic sea-ice – ocean – life coupled processes and therefore keep the main message of the original title.

The chronology needs to be improved. Fitting a third-order polynomial function through the dates is clearly not ideal. Some radiocarbon date probability distributions fall entirely outside the final age-depth curve (e.g. UBA-21627, UBA-21482, UBA-21639). A Bayesian approach to the age modelling would be preferred (using Oxcal or Bacon), but even a linear interpolation between the median calibrated ages would give a more reliable result than the one presented.

Reply: Yes, we performed a revision of the age-depth modelling by using a Bayesian approach (Bacon v2.2) as suggested (line 154-163). New Figures (4, 5, 6) with the new age model are given in the revised version of the manuscript.

The modelled sedimentation rate is then used to argue for a slow and continuous advance of the ice sheet towards the shelf break (lines 183-184, lines 194-195). There is less than 1 date per 1000 years and the fitted curve is unrealistically smoothed. Based on Figure 4, the highest sedimentation rates seem to actually occur just after 20 ka, but this is where 2 dates are falling off the age model curve. This later period of maximum sed rate would also match nicely with the presented IRD and Zr/Al data, as written in lines 235-236. Sometimes the modelled age-depth curve is used, sometimes the linear interpolation.

Reply: Based on the Bayesian approach, we have calculated the sedimentation rates for each time step and compare the results with a linear sedimentation rate based age model between all AMS¹⁴C fixpoints as shown in the new Figure 5. Both approaches complement each other and convincingly illustrate the gradual build-up of the ice sheet with a first maximum around 23 ka (first shelf edge glaciation) and a second maximum during break up around 19 ka.

The two studied cores are matched up using XRF Ca records and radiocarbon dates are subsequently moved from one depth scale to the other. This method should be described in more detail. In Table 2, please show both the original and converted core depths. The detailed description together with Figure 3 might be better placed in supplementary information.

Reply: We use the statistical software programme AnalySeries to correlate the different depth-scales between both cores. We have included this with a new reference (Paillard et al. 1996) and prefer to keep it in the main text as we consider it to be a key point. Table 2 now shows both the original and converted core depths.

The IRD pulse at 19 ka shows a jump from 0-5% to ca. 25%. All other measured variables seem unaffected by this change. If suddenly a quarter of the material is derived from ice-rafting, this should be visible as a slight drop in concentrations of forams, dinosterol, IP25 per gram. How come this doesn't affect any other proxies?

Reply: The biomarkers are associated with the total organic matter in the fine fraction, which is still 70% of the total. We show the data as they are and won't provide any speculations.

Figures 2, 5, and 6 all show data from the studied sediment core, but they are not all showing the same interval. I suggest to add a depth scale to figures 5 and 6 besides the age – axis. In the top of Figure 2, the sand content increases dramatically, but this is not shown in figure 5 or 6? Figures 5 and 6 show nearly the same time interval but not entirely, I suggest the authors correct this.

Reply: Yes, the sandy-rich, core top as indicated in Figure 2 and described in the “Lithology” chapter has not been sampled for this study. This is now indicated in the main text (lines 144-147). Figure 5 presents the entire core as illustrated in Figure 2 except the topmost, sandy-rich part. The top age of Figure 6 is now identical to Figure 5. However, the time scale in Figure 6 focusses on LGM and initial deglaciation only to better compare with other Nordic Seas/North Atlantic records.

The comparison of data in Figure 6 forms the basis for the interpretation of large-scale polynyas in front of the SBIS. It is not clear from the figure and the caption which curve belongs to which core (e.g. the dotted line of IP25). Curve (a) shows oxygen isotopes from two cores, with about 1 ka difference in timing of the meltwater pulses, but this is not discussed. The same time lag can be seen in the IP25 records. Curve (b) doesn't show a comparison of IRD between two records but rather uses Zr/Al, why? Curves (c) and (d) in Figure 6 seem to only be included to support a very short statement in lines 283-284. This should be expanded. Neither of these sites are shown on a map.

Reply 1: We have changed the caption of Figure 6 (see below) to provide easy identification of the different records.

Figure 6: Sea ice and ice sheet dynamics along the western/northern Barents Sea margin during the LGM. (a) Planktic $\delta^{18}\text{O}$ (in ‰) and (b) ice-rafted debris (% >250 μm) records from the southwestern (this study: in green) and northern (PS2138-1: in black) Barents Sea margin ⁶⁰. (c) IP₂₅ in sediments off northwestern Svalbard (MSM05/5-712-2) (dotted line) ⁵¹ and southwestern Barents Sea (green) (this study). See locations of the cores in Figures 1 and 7. (d) Benthic foraminifera $\delta^{13}\text{C}$ in the Norwegian Sea (PS1243, black) ¹⁵ and the North Atlantic (ODP Site 980, gray) ⁶⁷. Timing for ventilation changes in the Nordic Seas and AMOC slow down are indicated by arrows. See given references for exact location of these cores. For captions of coloured rectangles, see Figure 5.

Reply 2. The discrepancy of ca. 1 ka of the meltwater onset and IP_{25} difference between the cores from SW Barents Sea and western/northern Svalbard is discussed in the revised version. (lines 256-276).

Reply 3: We now show the IRD records of both cores in Figure 6. We have omitted the Zr/Al record.

Reply 4: We have expanded the discussion on the benthic isotope record from the Norwegian Sea and North Atlantic in the revised version of the manuscript (lines 229-235, lines 285-297).

Several of the cores in Figure 7 are not mentioned in the text.

Reply: We have re-phrased the caption of Figure 7 and provide better reference to the main text. In addition. We now refer to Supplementary Table 1 and provide the original references for the source data.

Line 74 and 314: correct oases to oasis.

Reply: We have controlled the use of oasis and oases with our native English co-authors and corrected accordingly.

Figure 1A: In the legend, the bathymetry color scale is a continuous gradient of shades of blue. The map itself however doesn't follow this color scheme: the deeper sites are much more purple.

Reply: Yes, correct. We have limited the color scale to less than 1000 m as indicated in Figure 1A to highlight the different contours from the upper continental slope to the shallow banks on the Barents Sea shelf.

Figure 1A: Add label for the Bear Island Trough

Reply: corrected

Line 108: Core name ends in -01PC on the map, but -1PC in the text. Use a consistent name.

Reply: corrected

Line 109: Water depth of core site: is it 990m (main text) or 949m (Table 1)?

Reply: corrected

Lines 144-145: remove "that operates with a standard =0)". This is not entirely correct. The difference between the atmospheric and marine curve is approx. 400 years on average but this is not the same as R . Simply state that $\Delta R = 0$ years is enough.

Reply: corrected

Figure 5 and 6: Describe the red and blue shading in the plot. Red is LGM? What is blue?

Reply: yes, light reddish colored rectangle indicates the LGM according to Clark et al. (2009). The bluish rectangle marks the freshwater release from the Barents Sea ice sheet. This has been changed in the Figure captions.

Figure 5b: Sedimentation rates based on the age model of Figure 4. Why are there so much high frequency wiggles on this curve? The age model is a smooth curve so why would there be so much small variability in the sedimentation rate?

Reply: To clarify, the sedimentation rate model is not smoothed, but calculated between each age-depth fixpoints provided by the Bayesian age-depth modelling approach using the Bacon v2.2. software.

Figure 5e: The left of this curve seems cut off. Is there a data point before 16 ka which is not shown?

Reply: The core has been sampled at various depths due to limited material available for the different sedimentological and geochemical analyses. Supplementary Table 2 provides all produced data, however, for consistency purpose we present the data from ~15.5 ka BP onwards in Figure 5.

Revisions recommended by reviewer 2

My main concerns regards the lack of consideration of transport of the material containing the biomarkers. These are predominantly in the fine fraction which is highly mobile and can be transported by currents 100s to 1000s of km (eg McCave et al 2001, Science and refs therein; Filippova et al 2016 Paleocyanography), with the finer clay fraction being transported across large basins (eg Fagel and Mattielli 2011 Paleocyanography). The study site here is one that (as stated by the authors) is strongly influenced by the northward transport of the Norwegian Atlantic Current, which has been shown to be capable of sorting and transporting coarse silts (Tegzes et al 2017, The Holocene) and building sediment drifts. How to the authors of this study constrain the provenance of the sediment size fraction containing the biomarkers and how do they rule out that it is not an allochthonous signal? Their record could simply be one of advected biomarkers from further south. Complementary data from sand size proxies such as foraminifera helps provide support for their interpretation of in situ seasonal sea-ice cover, but the authors must do a lot more to inform the reader of this major weakness in their IP₂₅ evidence, and it would have been useful if they presented a case to rule out advection artefacts on their IP₂₅ record.

Reply: Yes, indeed, we have been discussing the problem of lateral advection of biomarkers and whether we can exclude the possibility that our IP₂₅ variability is the result of bottom water dynamics. To address this problem, we now refer to a recent publication by our team (Közeoglu et al. 2018) in *Geochimica et Cosmochimica Acta*. For the quantitative assessment of sea ice conditions in the Norwegian/Barents Sea region, we compiled a large

data set of IP_{25} measured on surface sediment samples (0-1 cm). The samples have been collected in various settings within high-accumulation fjords to continental slope and shelf areas influenced by high bottom current activity. The map below shows the presence/absence of IP_{25} in modern sediments (white vs. blue circles) as well as the difference to the mean concentration in all IP_{25} containing sediments (large blue circles are above the mean IP_{25} value; small blue circles are below the mean IP_{25} value). From the map it is obvious that strong bottom currents along-slope or down-slope have no influence on absence or presence of IP_{25} in the sediments. The abundance is strictly controlled by the present-day sea ice margin implying that vertical transport dominates the preservation of the signal in the sediments. Second, smaller blue circles (and thus IP_{25} concentration close to the mean IP_{25} in all samples) largely occur at the winter sea ice limit suggesting that only the vertical transport of “marine snow” is the main carrier of the preserved signal in the sediments. This observation let us conclude that the IP_{25} record at the core site GS14-190-01PC is not affected by lateral transport processes, particularly not during the LGM and deglaciation where bottom water current activity was obviously much less intense than today. We provide reference to this article in the revised version of the manuscript. We also include this figure in the revised manuscript as supplementary information.

Figure: Map showing the absence and presence of IP_{25} in surface samples (0-1 cm) in the Norwegian and Barents Sea as a baseline for the Közeoglu et al. (2018) paper in *Geochimica et Cosmochimica Acta*. The location of piston core GS14-190-01PC is marked.

Lateral transport by bottom currents alongslope or downslope do not influence the surface pattern of IP₂₅ concentration. Its concentration is primarily controlled by the presence or absence of sea ice and vertical transport of particulate organic material through the water column.

I am supportive of the work, but am not convinced about how this represents a major advance from what the community already knew, as shown by the useful references to earlier work provided, which reach similar conclusions. Please could the authors highlight specifically how this is an advance from earlier work – more than just listing earlier evidence as being speculative whereas theirs is definitive. Thanks.

Reply: Yes, we highlight the advances from earlier work in the introduction (line 97-103) and the discussion (lines 224-235). It is important to stress that the current view of pulse-like heat flow through variable Atlantic water inflow was, according to our study, not the main controlling factor for variable sea ice dynamics and thus seasonally sea-ice free conditions in the Nordic Seas, and specifically the eastern Fram Strait. Instead, our data indicate that a combination of katabatic winds blowing off the NW Eurasian ice sheets, pushing the approaching sea ice margin westwards and northwards, and irregular upwelling of sub-surface Atlantic water created polynyal activity that produced an ice-free corridor in front of the ice sheets during the Last Glacial Maximum.

The assumptions made in the age model construction are likely flawed – specifically the use of a 400 yr reservoir age. This is likely to be variable and 400yr seems far too low. Mixed benthic/planktics were used (including *N pachyderma*?), yet studies using these signal-carriers from Nordic Seas and N Atlantic show reservoir ages of up to 2000 years (Voelker et al 1998, Waelbroeck et al 2001, Sarnthein et al 2001, Stern and Lisiecki 2013, Thornalley et al 2011 and 2015, Ezat et al 2017). Greater consideration of variable reservoir ages should be made, including error estimates and at least mentioning the possibility of much higher reservoir ages in the text, and briefly saying how this would shift the timing of events to younger calendar ages.

Reply: Yes, we agree and added a paragraph (line 154-163) why we have chosen a local reservoir age of $\Delta R = 0$. We are of course aware of large uncertainties in reservoir ages in the Nordic Seas during the LGM and early deglaciation and have inserted more references to this issue. We use a local reservoir age $\Delta R = 0$ here to enable a more direct comparison to other key records from the North Atlantic and Nordic Seas. Age models published by McManus (2004, Nature), Thornalley (2015, Science) and Müller and Stein (2014, EPSL) and from which we show key proxy records (benthic $\delta^{13}\text{C}$ and IP₂₅) in Figure 6 operate either with $\Delta R = 0$ (McManus, Thornalley) or $\Delta R = 40$ (Müller and Stein). Applying the same ΔR value in our study allows us to transfer our observations on a same time line to other key proxy records for the LGM and deglaciations and discuss changes more directly. This is now better indicated in the Chronology chapter. Of course, we provide the raw data as well in the Supplementary Table 2 for any further age-depth modelling improvements.

The study only uses concentration per gram of sediment, which can be susceptible to changes in the basin wide input of terrigenous material. Usually the effects of changes in sediment flux and focussing are constrained using Th-normalization to better understand

sediment fluxes. It does not seem like the major results would be countered by this effect (because IP₂₅ conc increases at a time when sed rate also increases; and fortunately the authors have not then taken the next step of converted to a flux per year which would have been erroneous without Th-normalization), however it significantly limits the interpretation of downcore trends other than the major increase and decrease, since transient decreases in concentration may be due to episodic basin input of terrigenous material.

Reply: A slight correction here. Indeed, in Figure 5 we show a normalization on gram sediment, however, for comparison purposes to another key IP₂₅ record from western Svalbard margin (Müller and Stein 2014), we normalize to TOC in Figure 6. We discuss primarily the major trends. Clear increase in IP₂₅ and dinosterol as well as planktic forams (all normalized to gram Sed.) during the LGM as evidence for a polynyal activity, and complete collapse during freshwater release. Transient decrease is discussed for the TOC normalized IP₂₅ record to provide an explanation for the discrepancy to the record from NW Svalbard during the initial shelf edge deglaciation. As the reviewer correctly pointed out, calculations of flux rates without Th-normalization would be erroneous, we rely on the interpretation of these large trends in our proxy records. No further changes have been made in the text.

L55 (and later text) – I would be careful overstating the role of brines in AABW formation – important, but perhaps it is useful to mention/reference the other processes, citing eg Toggweiler and Samuels 1995 (J Phys. Oceanogr., 25, 1980)

Reply: Thanks for this. We have modified the text to be more circumspect about the role of brines in AABW formation. We have not included any additional reference here due to limited space for references.

L276 – please reword to avoid phrases like “leaves little doubt that”. It sounds like you are trying too hard to convince the reader by saying these phrases, rather than simply providing the evidence to support your case. Moreover, previous studies (eg Dokken and Jansen 1999, Nature) have provided similar evidence for d18O anomalies that are as large as the planktic d18O presented here, so it is unclear why this is such a major advance and now leaves us in no doubt (if we were before). I would instead just say that the large planktic d18O provides further support for the role of SBIS in meltwater input at this time.

Reply: We do not agree here. The Dokken & Jansen (1999) core is from the Mid-Norwegian margin and does not reflect the collapse of the SW Barents Sea ice sheet, but rather a combination of Barents Sea and the Fennoscandian ice sheet (Figure 2). The reviewer asks for major advances of this study. Yes, indeed, this is a major advance. To retrieve an undisturbed sequence from this part of the continental margin has so far been unsuccessful. Deglaciation ages (see Hughes 2015 in Boreas) of the SW Barents Sa shelf edge glaciation has thus far not been constrained. To increase the significance of our record even further, we have increased the resolution of the planktic $\delta^{18}\text{O}$ record and show a quite unique deglaciation pattern for this setting. We prefer to leave the paragraph, but have re-phrased it to clarify the significance of the new findings.

Figure: See Dokken and Jansen (1999) MD95-2010 core from the mid-Norwegian continental margin vs. GS14-190-01PC (red circle, this study).

L320 – please provide evidence or a reference for the high diversity of planktic, and especially benthic, diversity...none seems to be provided.

Reply: Yes, we inserted a new reference to support our statement: Rasmussen, T. L. *et al.* Paleooceanographic evolution of the SW Svalbard margin (76 degrees N) since 20,000 C-14 yr BP. *Quaternary Research* **67**, 100-114

Fig 7 – would be improved if the various evidence at each site was somehow shown or listed in the caption or elsewhere. As a reader it was not clear what core each site showed, without doing detailed searching of the text and linking up shorts comments in the text with the diagram – nicer if the authors can help the reader here.

Reply: We have adjusted the Figure caption including the references for each site (see also response to Reviewer 1). Each site shows the same characteristics as the core site GS14-190 from the SW Barents Sea. High sea ice diatom production is coupled to open marine phytoplankton and zooplankton production implying a very dynamic sea ice margin in front of the entire NW Eurasian ice sheet in an elsewhere permanently sea-ice covered Nordic Seas during the LGM.

Figure 7: Schematic illustration of the polynyal activity in front of the Svalbard-Barents Sea ice sheet during the LGM as reconstructed from 5 sediment proxy records (Table 1). Polynya activity is constrained by relatively high sea-ice diatom (IP₂₅), marine organic phytoplankton and calcareous zooplankton production in all displayed sediment cores (this study; Müller and Stein 2014, Müller *et al.* 2009, Stein *et al.* 2012, Knies and Stein 1998, and Xiao *et al.* 2015) supported by sub- surface/intermediate inflow of Atlantic-water derived waters (NAC) and katabatic winds blowing off the Svalbard-Barents Sea ice sheet. Coastal polynyas along the entire Svalbard-Barents Sea margin provided a constant source of moisture to sustain build-up of glacial ice, ventilation of deeper waters in the glacial

Nordic Seas, and remained a refuge for marine and higher trophic terrestrial life in a polar desert. Abbreviations are: Sv: Svalbard, SBIS: Svalbard-Barents Sea Ice Sheet, BIT: Bear Island Trough

Revisions recommended by reviewer 3

The study by Knies et al. deals with an important subject in marine glacial climate reconstructions, namely the crucial role of polynya systems as drivers on both marine biotic production and ocean circulation in a glaciated polar region. Overall I quite like the approach in general. But there are some shortcomings, which, I believe, can be dealt with by the authors during revision. One is to do with the fact wind forcing has on not only creating open water leads (polynyas) and enhancing local bioproductivity, but also causing upwelling that would inevitably lead to increased ocean circulation such as advection of Atlantic waters from subsurface-intermediate depths with shallow overturning at probably also an intermediate water depths. Such situations have been suggested/interpreted before by various authors in studies from continental arctic margins and open polar ocean (Noegaard-Pedersen et al, Taldenkova et al., Bauch et al....). Although the authors mention it I believe this topic deserves some better strengthening and further elaborating... especially because it is still consensus among many of the "paleocommunity" that ocean circulation in the polar ocean was very sluggish and insignificant.

Reply: Yes, we agree that sub-surface inflow of Atlantic-water in addition to the seaward blowing winds is decisive for the dynamics of the marginal ice zone in front of the ice sheet. We think that these coupled processes are adequately addressed in the revised manuscript (e.g. lines 97-103, 224-235, 321-327).

I support the idea that the onset of collapse of the Barents ice caps and shelves caused a dramatic change in the habitat of marine plankton biota, at least in the proximity of thick glacier ice. To what areal extent, however, remains rather speculative as the center of productions could have easily shifted elsewhere like into the central parts of the Nordic seas (e.g., Iceland Sea) where the meltwater lid was very likely of lesser thickness.

Reply: We think it is important to mention here that with the observation of IP₂₅ close to the SW Barents Sea margin in this study, it is very likely that the Nordic Seas were largely (if not perennial) sea-ice covered during the LGM, limiting any high productivity away from the site of polynya formation. Summer meltwater lids in surface waters and sea-ice coverage throughout the remaining time of the year have been suggested earlier for the Fram Strait and Iceland Sea (e.g. Bauch et al. 2001) likely excluding the possibility of a shift of the high production cell towards the central Nordic Seas. This is now better illustrated in the revised version of the manuscript.

However, such a scenario as the reviewer describes here cannot not be fully ruled out and would be very important to test. More IP₂₅/HBI records from key sites in the Iceland and Greenland Seas are therefore required to provide some ultimate inferences.

And also, while the Barents Sea ice sheet was quickly gone during early deglaciation, I wonder if there is more supporting evidence (IP₂₅ and other proxies) of a continuation of the polynya processes described by the authors until 15ka in other areas where the ice sheet

(and katabatic winds) remained close by in the hinterland for a substantial longer time during deglaciation. The region west of Spitsbergen with its narrow shelf is an adequate area to look into...and the cores are listed here.

Reply: The illustration of the polynya in Figure 7 is based on proxy data in the given cores (similar to the observation made along the southwestern Barents Sea margin (core GS14-190). We have improved the caption of Figure 7 to better illustrate the supporting evidence in these records for polynyal activity based on IP_{25} and other phytoplankton proxies. It is now evident that the entire margin in front of the NW Eurasian ice sheet, from the southwestern Barents Sea, western Svalbard, to the northern Barents Sea margin was influenced by coastal polynyas. The revised Figure 7 should now make this point clear.

REVIEWERS' COMMENTS:

Reviewer #1 (Remarks to the Author):

The authors have submitted a thoroughly revised manuscript addressing the recommendations of all reviewers and I can now recommend to accept the paper for publication.

Reviewer #2 (Remarks to the Author):

I thank the authors for addressing the comments made during the review. I am happy with their reply and changes and the revised manuscript is to be commended. I was especially impressed by the very nice core top data figure showing the presence of IP25 versus today's sea-ice edge...nice data and thanks for showing. I am happy with the added mention of age model uncertainty and significance of the planktic d18O peak.

This will be a useful and important contribution to the literature of this region and has broader lessons/significance. It is well written with clear figures.

My only remaining suggestion at this point is that I would agree with Reviewer 1 that the title is a bit broad and could be improved. How about (and is their space for) something like:

"Arctic polynyas and their role in preconditioning life during the Last Glacial Maximum: evidence from Svalbard"

Reviewer #3 (Remarks to the Author):

For the revision I think the authors did a good job in accommodating most of the reviewers' comments adequately. One minor thing that came to my mind, and which could be easily added, is the notion on the polynya system and how it facilitated a local benthic "oasis" (line 315 - 318). That situation probably not just changed the benthic community by showing *C. neoteretis* but also affected planktic life that is otherwise often related/interpreted to direct Atlantic water inflow at the surface. Smaller-sized planktonic specimens of *T. quinqueloba* have been noted to occur in the study area and around during the last glacial (Dokken & Hald 1996; Bauch et al. 2001; Telesiniski et al. 2015), and I would like to see the author appreciate this fact by adding the fact here. Otherwise I am quite happy with the ms and look forward to seeing the manuscript published asap.

REVIEWERS' COMMENTS:

Reviewer #1:

The authors have submitted a thoroughly revised manuscript addressing the recommendations of all reviewers and I can now recommend to accept the paper for publication.

Reply: Thank you!

Reviewer #2:

I thank the authors for addressing the comments made during the review. I am happy with their reply and changes and the revised manuscript is to be commended. I was especially impressed by the very nice core top data figure showing the presence of IP25 versus today's sea-ice edge...nice data and thanks for showing. I am happy with the added mention of age model uncertainty and significance of the planktic d18O peak.

This will be a useful and important contribution to the literature of this region and has broader lessons/significance. It is well written with clear figures.

My only remaining suggestion at this point is that I would agree with Reviewer 1 that the title is a bit broad and could be improved. How about (and is their space for) something like:

"Arctic polynyas and their role in preconditioning life during the Last Glacial Maximum: evidence from Svalbard"

Reply: After discussion with the editor Lewis Collins, we propose a new title:

“Nordic Seas polynyas and their role in preconditioning life during the Last Glacial Maximum”

Reviewer #3:

For the revision I think the authors did a good job in accommodating most of the reviewers' comments adequately. One minor thing that came to my mind, and which could be easily added, is the notion on the polynya system and how it facilitated a local benthic "oasis" (line 315 - 318). That situation probably not just changed the benthic community by showing *C. neoteretis* but also affected planktic life that is otherwise often related/interpreted to direct Atlantic water inflow at the surface. Smaller-sized planktonic specimens of *T. quinqueloba* have been noted to occur in the study area and around during the last glacial (Dokken & Hald 1996; Bauch et al. 2001; Telesiniski et al. 2015), and I would like to see the author appreciate this fact by adding the fact here. Otherwise I am quite happy with the ms and look forward to seeing the manuscript published asap.

Reply: We changed the sentence in lines 314-318 to “Indeed, previous reports of unusually high abundances of the benthic foraminifer *Cassidulina neoteretis* and smaller-sized

planktonic specimens of *Turborotalita quinqueloba* in LGM sediments off western Svalbard^{14,40,48,51} indicate the existence of a productive benthic oasis in an otherwise glacial desert at this time, with nutrient supply and organic matter production in a marginal ice zone supported by upwelling of nutrient-rich Atlantic-derived water masses.”